

# Simulating more realistic predation threat using attack playbacks

Mukta Watve[1], Sebastian Prati[2] and Barbara Taborsky[1]

[1] Division of Behavioural Ecology, Institute of Ecology and Evolution, University of Bern, Bern, Switzerland
[2] Department of Arctic and Marine Biology, Faculty of Biosciences, Fisheries and Economics, UiT The Arctic University of Norway, Tromsø, Norway

## ABSTRACT

Use of virtual proxies of live animals are rapidly gaining ground in studies of animal behaviour. Such proxies help to reduce the number of live experimental animals needed to stimulate the behaviour of experimental individuals and to increase standardisation. However, using too simplistic proxies may fail to induce a desired effect and/or lead to quick habituation. For instance, in a predation context, prey often employ multimodal cues to detect predators or use specific aspects of predator behaviour to assess threat. In a live interaction, predator and prey often show behaviours directed towards each other, which are absent in virtual proxies. Here we compared the effectiveness of chemical and visual predator cues in the cooperatively breeding cichlid *Neolamprologus pulcher*, a species in which predation pressure has been the evolutionary driver of its sociality. We created playbacks of predators simulating an attack and tested their effectiveness in comparison to a playback showing regular activity and to a live predator. We further compared the effectiveness of predator odour and conspecific skin extracts on behaviours directed towards a predator playback. Regular playbacks of calmly swimming predators were less effective than live predators in stimulating a focal individual's aggression and attention. However, playbacks mimicking an attacking predator induced responses much like a live predator. Chemical cues did not affect predator directed behaviour.

## INTRODUCTION

Predation risk is thought to play an important role in determining life-history trajectories, the social system and behaviours among animals (*Holling, 1965*; *Tollrian, 1995*; *Stearns, 2000*; *Heg et al., 2004*; *Preisser & Bolnick, 2008*; *Walsh & Reznick, 2008*; *Groenewoud et al., 2016*; *Fischer et al., 2017a*). While an obvious way of testing the impacts of predation risk is to manipulate the number of predators and/or to present predators to potential prey (e.g., *Chiverton, 1986*; *Fontaine & Martin, 2006*), such experiments are not always logistically or ethically feasible. An alternative approach is to use 'virtual proxies' on captive prey, which reduces the ethical implications of subjecting such prey to live predators and removes the logistical and ethical issues of manipulating predators. Many proxies such as decoys (*Arroyo, Mougeot & Bretagnolle, 2001*; *Li et al., 2014*; *Catano et al., 2016*; *Griesser & Suzuki, 2017*; *Morales, Lucas & Velando, 2018*), screen presentations (*Johnson & Basolo, 2003*;

Corresponding author
Mukta Watve,
mukta.watve@iee.unibe.ch

*Fischer et al., 2014*; *Scherer, Godin & Schuett, 2017*; *Watve & Taborsky, 2019*) or kairomones and alarm cues (reviewed in *Ferrari, Wisenden & Chivers, 2010*) have been used successfully so far. Nevertheless, the utility of virtual proxies rests with their ability to replicate the natural predator–prey encounter. Prey may use multiple cues to detect a predator (*Gonzálvez & Rodríguez-Gironés, 2013*) or multiple cues may have a different effect than a single cue (*Hartman & Abrahams, 2000*; *Hickman, Stone & Mathis, 2004*; *Martin et al., 2010*; *Ward & Mehner, 2010*). Further, prey may use specific features of the predator's behaviour or orientation to assess the immediate risk and fine-tune their behaviour (*Kent et al., 2019*). Most importantly, live predators and prey will naturally interact, an element inherently lacking in the use of proxies. Despite this, such interactions are rarely incorporated into experiments using virtual proxies (but see *Hartman & Abrahams, 2000*)—increasing the chances that prey will quickly habituate or even ignore the stimulus (*Holomuzki & Hatchett, 1994*; *Raderschall, Magrath & Hemmi, 2011*). To minimize these shortcomings, it is necessary to ensure that virtual proxies (i) stimulate the relevant sensory modalities and (ii) provide salient features of natural predators eliciting ecologically relevant prey responses.

In aquatic systems, the role of visual as well as chemical cues in predator detection and response is well studied. Chemical cues released from damaged tissues of injured conspecifics ('alarm cues') play an important role in detection of predators in many species and are used in experimental manipulation (e.g., *Brown & Smith, 1998*; *Hartman & Abrahams, 2000*; *Brown, 2003*; *McCormick & Manassa, 2008*; *Ferrari, Wisenden & Chivers, 2010*; *Manassa & McCormick, 2012*; *Moscicki & Hurd, 2017*; *Lucon-Xiccato, 2019*). For example, skin extract from conspecifics administered along with predator odour helps fishery-reared rainbow trouts to recognise predators (*Brown & Smith, 1998*). Similarly, the use of visual proxies in aquatic systems is also well-established (e.g., *Johnson & Basolo, 2003*; *Gerlai, 2013*; *Fischer et al., 2014*; *O'Connor et al., 2015*; *Dieng et al., 2017*). While most studies report a successful use of proxies in inducing predicted anti-predator response, none have compared their effectiveness to live predator presentations.

*Neolamprologus pulcher* is a cooperatively breeding cichlid from Lake Tanganyika that lives in groups of up to 25 individuals (*Taborsky, 2016*). The social structure of *N. pulcher* is driven primarily by predation pressure (*Groenewoud et al., 2016*) and thus, manipulation of predation threat has been often employed to study their social dynamics (e.g., *Heg & Taborsky, 2010*; *Fischer et al., 2017a*; *Watve & Taborsky, 2019*). Turbidity and water quality in the lake differs between populations (*Plisnier, 2002*) and between seasons (*Plisnier et al., 1999*). Therefore, it is expected that *N. pulcher* would rely on both visual and chemical cues flexibly. *Lepidiolamprologus elongatus* is a common and ferocious predator of *N. pulcher* that is most often used in manipulation experiments in field as well as laboratory. It has been shown that *N. pulcher* can successfully distinguish images of *L. elongatus* from herbivorous cichlids of the same size and respond accordingly (*Fischer et al., 2014*). Chemical cues from *L. elongatus* alone can also induce appropriate anti-predator responses in very young fry of *N. pulcher* (*Fischer et al., 2017a*). Predator recognition in N. pulcher seems to be innate, as both these studies observed anti-predator response by naïve individuals. One study showed an unexpected reduction in response towards visual predator cues in the presence

of conspecific skin extract on the anti-predator response of *N. pulcher* (*O'Connor et al., 2015*). It is, however, unknown if and how a combination of both predator odour and conspecific skin extract influences anti-predator response of *N. pulcher*.

Here we manipulated visual proxies to include prey-directed aggression by the predator *L. elongatus* to simulate a live interaction. We further tested the effect of predator odour and conspecific skin extract independently and in combination with each other on anti-predator responses of *N. pulcher*. We expected playbacks simulating directed attacks to induce responses more similar to a live predator but for lateral playbacks of normally swimming predators to induce lower responses than a live predator. We also predicted that *N. pulcher* will show a heightened response towards the predator visuals when predator odour was administered in conjunction with conspecific skin extract.

## METHODS

### Experimental animals and housing conditions

The experiments were conducted at the Ethologische Station Hasli, University of Bern, Switzerland, under license BE-74/15 of Veterinary office of Kanton Bern. All experimental animals were obtained from the University of Bern laboratory stock, maintained in a 13:11 h light-dark cycle at 26 °C $\pm$ 1 °C mimicking the conditions of Lake Tanganyika. All focal *N. pulcher* were predator-naïve adults (size range 40–45 mm) reared in the laboratory for 5 generations. All predators were first generation offspring of wild caught fish and were in the size range of 80–140 mm. The experimental 20-litre tanks (40 cm × 25 cm × 20 cm) were lined with 2 cm layer of sand and aerated by air stones. They were visually divided in three zones marked by pen on the front screen of each tank. The zone closest to the stimulus presentation was designated as 'zone of interaction', where the focal individual actively engaged in inspection of and defence against the stimulus. The farthest zone was considered as 'safe zone' and the time spent in this zone was taken as a measure of avoidance (Fig. 1). Each focal was tested in a separate experimental tank and was allowed to acclimatise overnight before testing. During acclimatisation, focal individuals were provided with ceramic pot halves as shelters, which were removed half an hour before each behavioural recording to avoid obstruction of observations. Visual playbacks were displayed on a 25 cm × 15 cm LED tablet screen ('HP ElitePad 900', resolution: 59 dot/cm) in both the experiments.

### Experiment 1: Regular playbacks vs playbacks of attacks

Individual *L. elongatus* ($N = 2$) were housed in individual 20 L tanks overnight and then filmed swimming calmly for 5 min (hereafter 'lateral playbacks'). The same individuals were then filmed through a one-way mirror, as they attacked their own reflection (hereafter 'frontal playbacks'). The mirror and camera were placed along the breadth of the filming tank, elevated to align with the level of sand (2 cm from bottom). The lens of the camera was higher by a further 3–4 cm. During presentations, the tablet was again slightly elevated, such that the apparent level of sand on screen matched the level of sand in the experimental tank, such that the visuals on screen appeared similar to a real adjacent tank of same dimensions. Frontal playbacks also lasted 5 min and were edited in Adobe Premier Pro such that the
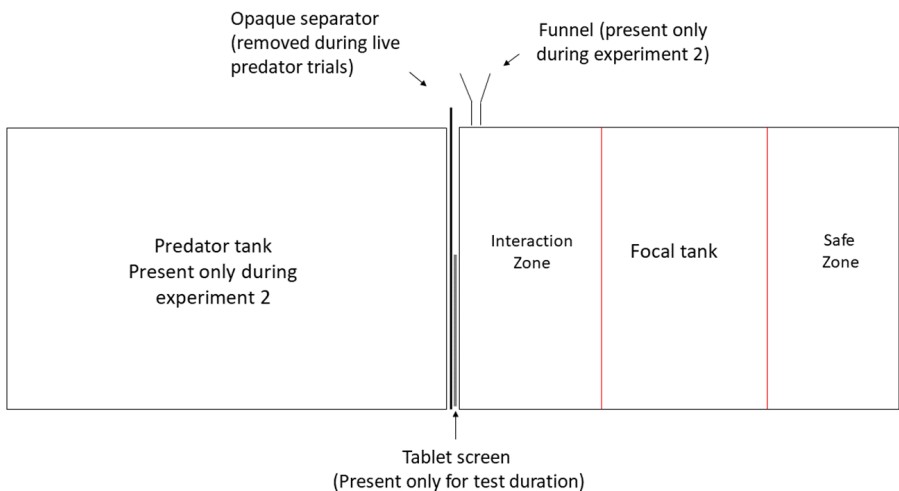

**Figure 1** **Experimental set up.** Experimental manipulation of visual and olfactory cues. Predator tank was present only during Experiment 1 and was separated from the focal tank with an opaque partition at all times except during live-predator trials. Olfactory cues were administered close to the presentation through a funnel during Experiment 2.

gap between any two attacks did not exceed 20 s. While it is theoretically possible that the responses towards a real prey and a conspecific mirror image differ in frequency and duration of aggressive displays (such as operculum spread, fin spread), frontal playback videos were edited to almost exclusively showed overt attacks (fast approach towards the target followed by biting), which at least for the human observer are indistinguishable from a targeted attack at prey. The filming tanks with predators were placed between two experimental tanks such that the same individual *L. elongatus* also acted as live stimulus. The predator and experimental tanks were separated by an opaque partition which was removed only during live presentation trials. A 5-minute long visual of an unoccupied filming tank was presented as blank control. To ensure that the focal fish were not simply responding to movement, an off-white square (approximately matching the colour of *L. elongatus*) of 5 cm × 5 cm was added on the background of the blank control and animated in PowerPoint to move randomly along the screen (hereafter 'moving square'). Each focal *N. pulcher* ($N = 16$) was exposed each to five presentations in a randomised order, with a gap of at least 3 h in between: (i) empty tank, (ii) moving square, (iii) calmly swimming predator, (iv) attacking predator, (v) live predator. The response of the focal towards the presentation was recorded for 5 min (see below).

## Experiment 2: Chemical cues

In this experiment we tested the response of *N. pulcher* to chemical cues from *L. elongatus*, skin extract from a conspecific and a combination of the two, compared to a control cue of water collected from the focal's own tank. To obtain predator odour, individual *L. elongatus* ($N = 3$) were housed in individual 60 L tanks (60 cm × 35 cm × 30 cm) for 48 h, fed once a day with standard cichlid flake food. They were not fed for another 24 h after which 500 mL water was collected from the tank and stored in −20C. Conspecific skin extract

was prepared as per *Brown & Smith (1998)*. Eleven *N. pulcher* in the size-range of 3.5–4.5 cm were sacrificed by stunning followed by severing the spinal cord. The skin was then separated, rinsed in distilled water to remove traces of other tissues and homogenised with equal volume distilled water (1 mL/1 cm$^2$) to release any possible alarm cues in skin cells. The homogenised mixture was filtered to remove debris and diluted 16 times. Aliquots of 12 mL were stored at $-20$ °C and diluted 10 times with water from an unoccupied experimental tank before use. For combining the two cues, 12 mL of skin extract was diluted with water from predator holding tanks instead. The final volume of all cues was maintained at 120 mL (0.6% tank volume). Focal *N. pulcher* ($N = 18$) were exposed to all four cues in randomised order with at least 3 h in between. This time window is sufficient for the cues to dissipate and the focal individuals to return to normal behaviour (*Watve & Taborsky, 2019*). All chemical cues were combined with visuals of a predator (from 9 different *L. elongatus* individuals) swimming calmly in a 20-L tank to facilitate 'predator directed' behaviour by the focal *N. pulcher*. Chemical cues were administered through a funnel at the beginning of the playback (see Fig. 1 for set-up). Responses of the focal fish were recorded for 5 min (see 'Behavioural recordings').

## Behavioural recordings

In both the experiments, the number of aggressive displays (head-down display, fin spread, operculum spread and fast approach towards the stimulus without attempting contact; see *Reyes-Contreras et al., 2019* for detailed ethogram) shown by the focal towards the stimulus display were counted as a measure of defence. If the focal individual was facing in straight line towards the screen while ceasing other activities (e.g., sand digging), it was considered as 'oriented towards the screen'. The amount of time a focal spent orienting towards the screen was used as a measure of attentiveness. Engagement with the stimulus was measured as the time spent in the zone closest to the predator presentation. Finally, avoidance or fear was measured as the amount of time spent in the safe zone (Fig. 1). Freezing, a common fear response of *N. pulcher*, was extremely rare and was therefore not included in the analysis. All observations were video recorded and later coded in Solomon Coder (András Petér, University of Budapest) with observer (SP) being blind to the type of cue.

## Statistical analysis

All analyses were conducted in R 3.5 (*R Core Team, 2017*). The frequencies of aggressive displays were analysed using generalised mixed models (GLMMs) with negative binomial distribution in package 'glmmTMB' (*Brooks et al., 2017*). The times spent being attentive to the screen and in safe zone were analysed using linear mixed models (LMMs) in package 'lme4' (*Bates et al., 2015*). Normality assumptions were tested by Shapiro–Wilk normality tests and by visually examining Quantile-Quantile (Q-Q) plots as well as histograms of residuals and plots of residuals against fitted values. Log and root transformations were employed to achieve normality when the model residuals violated normality assumptions. All models included cue as the main effect (five levels for experiment 1, four levels for experiment 2) and focal identity as a random effect. As there were only two predator individuals in experiment 1, they were included as a factor in initial models but had no

effect in any of the models. Removing predator identity decreased the AIC by at least 2 in all models and therefore, the final models only included cue as the main factor. The identity of the predator in visual displays ($N = 9$ different individuals) shown together with the chemical cues was included as a random effect in all models for experiment 2. In case of significant cue effects, pairwise comparisons between levels of the cue were carried out in package 'emmeans' (Lenth, 2018). P-values were adjusted for multiple testing using multivariate $t$ distribution (mvt) and only adjusted values are reported here.

## RESULTS

### Experiment 1

The number of aggressive displays towards the stimulus was determined by the type of presentation (GLMM, $\chi^2 = 23.41$, $p = 0.0001$; Fig. 2A). Pairwise comparisons between presentation types revealed that there was no difference between the frontal playbacks and live stimulus ($p = 0.13$). However, lateral playbacks induced much lower aggression than live stimuli ($p = 0.0008$). Further, there was also a significant difference between the control stimulus and frontal playbacks ($p = 0.02$), which was not the case for lateral playbacks ($p = 0.73$ and the moving square ($p = 0.98$; Table 1, Fig. 2). Also the time spent being attentive to the stimulus was affected by the type of presentation (LMM, $F = 4.14$, $p = 0.004$; Fig. 2B). Focals were more attentive towards the live stimulus as compared to frontal playbacks ($p = 0.02$), the moving square ($p = 0.004$) and the control stimulus ($p = 0.02$), but not compared to lateral playbacks ($p = 0.10$; Table 2). No difference was found between lateral and frontal playbacks ($p = 0.98$; Table 2). Further, the time spent in the interaction zone was also influenced by presentation type (LMM power transformed, $F = 5.34$, $p = 0.0009$; Fig. 2C). Fish spent much longer in the zone of interaction when encountering a live predator compared to a blank control ($p = 0.0007$) or a moving square ($p = 0.01$) and marginally longer when compared to a lateral display ($p = 0.09$). Attention time did not differ between frontal playbacks and live predator ($p = 0.46$) or between lateral and frontal playbacks ($p = 0.90$; Table 3). Time in the safe zone was not affected by the type of display (LMM log-transformed, $F = 1.29$, $p = 0.28$; Fig. 2D).

### Experiment 2

There was no effect of any chemical cue on the number of aggressive displays directed towards the screen (GLMM, $\chi^2 = 2.08$, $p = 0.55$; Fig. 3A), on the time spent attentive to the screen (LMM, $F = 1.15$, $p = 0.33$; Fig. 3B), or on the time spent in the safe zone (LMM, $F = 1.44$, $p = 0.24$; Fig. 3C) or in the interaction zone (LMM, $F = 0.36$, $p = 0.77$; Fig. 3D).

## DISCUSSION

Here we tested the influence of different chemical and visual predator cues on the response of *N. pulcher*. The type of visual playbacks effected only those variables related to an interaction between the focal fish and screen, but it did not affect avoidance behaviour. Focals spent similar time in the interaction zone and showed similar aggression towards live stimuli and frontal playbacks, however, the response towards lateral playbacks was

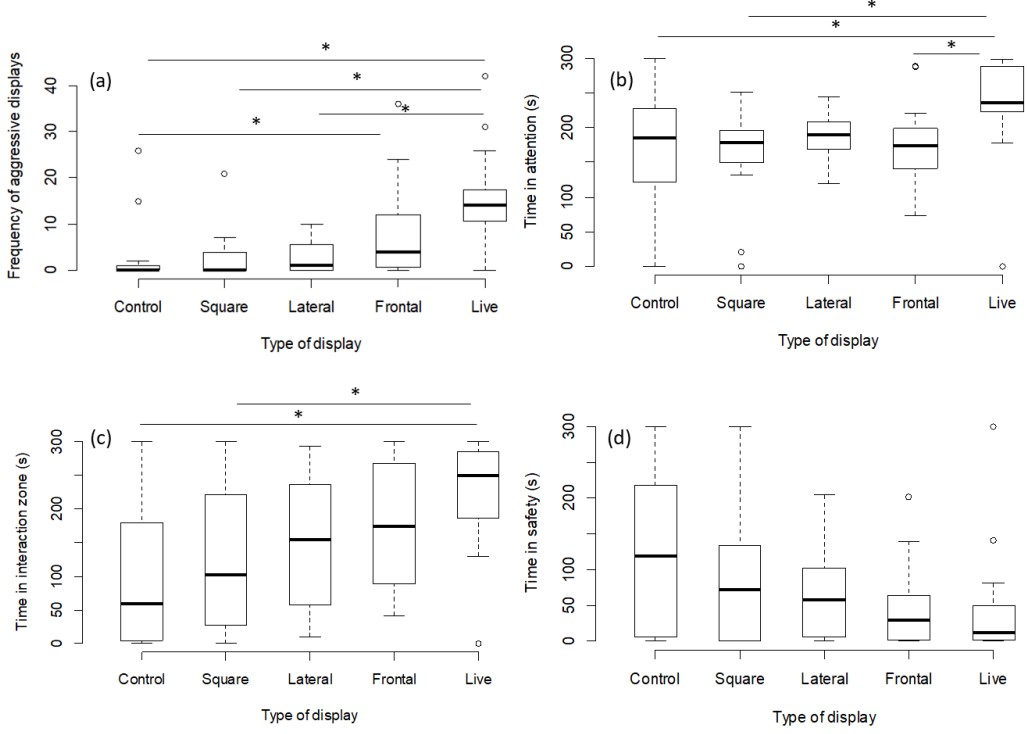

**Figure 2  Behavioural observations for experiment 1.** Effects of playback type on predator-directed behaviour ($N = 80$ observations of 16 fish). The graphs show the effects of lateral vs. frontal (attacking) predator playbacks in comparison to a live predator, a blank control and a moving square. Significant differences are indicated by asterisks.

**Table 1  Aggressive displays towards different types of predator presentations.** Pairwise comparisons between aggressive displays shown by focal *N. pulcher* ($N = 16$) towards different types visual predator presentations. All *p*-values presented are corrected for multiple comparisons using 'mvt' method. Significant *p*-values are marked with asterisks.

| Display | Estimate | SE | df | t | p |
|---|---|---|---|---|---|
| Control-Square | −0.26 | 0.55 | 73 | −0.47 | 0.98 |
| Control-Lateral | −0.70 | 0.57 | 73 | −1.22 | 0.73 |
| Control-Frontal | −1.77 | 0.57 | 73 | −3.08 | 0.02* |
| Control-Live | −2.98 | 0.62 | 73 | −4.76 | 0.0001* |
| Square-Lateral | −0.43 | 0.56 | 73 | −0.77 | 0.93 |
| Square-Frontal | −1.50 | 0.56 | 73 | −2.67 | 0.06. |
| Square-Live | −2.71 | 0.59 | 73 | −4.56 | 0.0002* |
| Lateral-Frontal | −1.06 | 0.52 | 73 | −2.05 | 0.25 |
| Lateral-Live | −2.28 | 0.54 | 73 | −4.17 | 0.0008* |
| Frontal-Live | −1.21 | 0.51 | 73 | −2.35 | 0.13 |

**Table 2 Attention towards different types of predator presentations.** Pairwise comparisons between time spent being attentive to different visual proxies by focal *N. pulcher* ($N = 16$). All *p*-values presented are corrected for multiple comparisons using 'mvt' method. Significant *p*-values are marked with asterisks.

| Display | Estimate | SE | df | t | P |
|---|---|---|---|---|---|
| Control-Square | 62.8 | 122 | 60 | 0.51 | 0.98 |
| Control-Lateral | −80.9 | 122 | 60 | −0.66 | 0.96 |
| Control-Frontal | −13.9 | 122 | 60 | −0.11 | 1.00 |
| Control-Live | −384.4 | 122 | 60 | −3.14 | 0.02* |
| Square-Lateral | −143.7 | 122 | 60 | −1.17 | 0.76 |
| Square-Frontal | −76.7 | 122 | 60 | −0.62 | 0.96 |
| Square-Live | −447.2 | 122 | 60 | −3.66 | 0.004* |
| Lateral-Frontal | 66.9 | 122 | 60 | 0.54 | 0.98 |
| Lateral-Live | −303.5 | 122 | 60 | −2.48 | 0.10 |
| Frontal-Live | −370.5 | 122 | 60 | −3.03 | 0.02* |

**Table 3 Time in interaction zone during different predator presentations.** Pairwise comparisons between time spent in interaction zone by focal *N. pulcher* ($N = 16$) during different visual proxies. All *p*-values presented are corrected for multiple comparisons using 'mvt' method. Significant *p*-values are marked with asterisks.

| Display | Estimate | SE | df | t | P |
|---|---|---|---|---|---|
| Control-Square | −26.1 | 30.7 | 60 | −0.85 | 0.91 |
| Control-Lateral | −52.1 | 30.7 | 60 | −1.70 | 0.44 |
| Control-Frontal | −79.2 | 30.7 | 60 | −2.58 | 0.08 |
| Control-Live | −130.0 | 30.7 | 60 | −4.24 | 0.0007* |
| Square-Lateral | −26.1 | 30.7 | 60 | −0.85 | 0.91 |
| Square-Frontal | −53.1 | 30.7 | 60 | −1.73 | 0.42 |
| Square-Live | −103.9 | 30.7 | 60 | −3.38 | 0.01* |
| Lateral-Frontal | −27.1 | 30.7 | 60 | −0.88 | 0.90 |
| Lateral-Live | −77.9 | 30.7 | 60 | −2.53 | 0.09 |
| Frontal-Live | −50.8 | 30.7 | 60 | −1.65 | 0.46 |

relatively weaker. Surprisingly, focals payed less attention towards the frontal playbacks than the live stimulus, whereas attention between the lateral playback and the live stimulus did not differ. The chemical cues did not affect the anti-predator behaviour of focals.

*N. pulcher* are sensitive to visual predator cues. For instance, they can distinguish heterospecifics, based simply on moving images (*Fischer et al., 2014*). However, images can fail to simulate a more realistic threat. Here we show that a playback of predator attack can induce a response similar to that shown towards a live predator. While prey species may recognize a predator based on images, they may not necessarily perceive them as threatening (*Belin et al., 2018*). Even for a moving visual, prey may only perceive the display as a threat if they include certain angles (*Kent et al., 2019*), postures (*Etting & Isbell, 2014*) or behaviours (*Su & Lim, 2017*) of the predator. Video playbacks have been used as predator cues in the past (*Rowland, 1999*; *Carlile, Peters & Evans, 2006*; *Butler, Magrath & Peters, 2017*). However, they lacked the element of prey-directed behaviour of the predator.

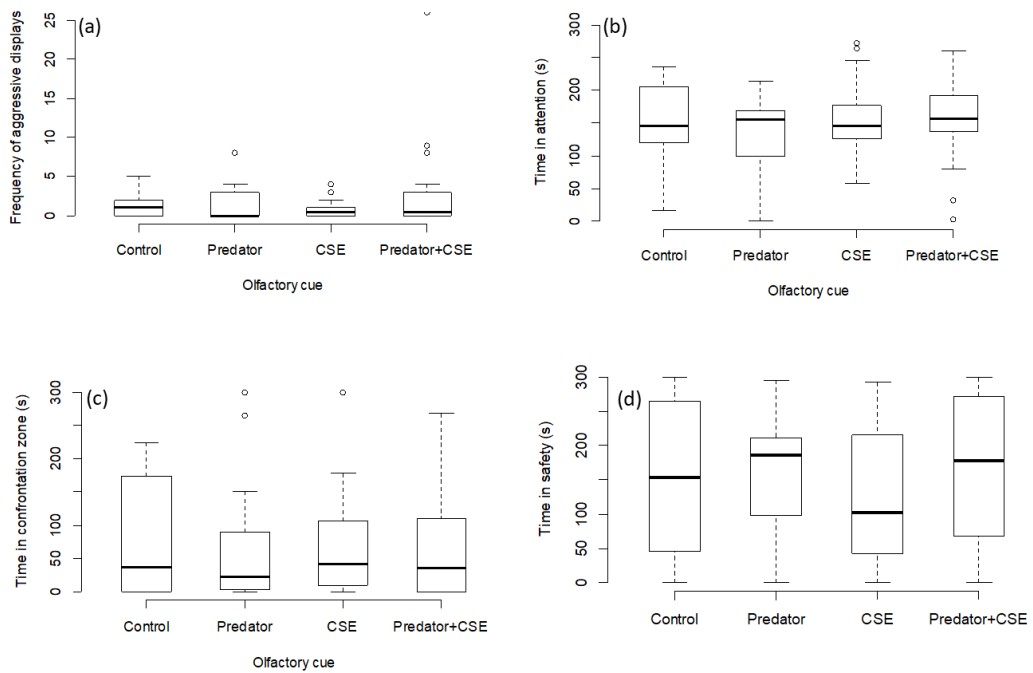

**Figure 3** **Results for experiment 2.** Manipulation of olfactory cues ($N = 72$ observations of 18 focal fish). The graphs show the effects of predator odour and conspecific skin extract (CSE) on the aggression, time spent being attentive to predator display and the times spent engaging with or avoiding the presentation. Significant differences are marked with asterisks.

We attempted to design visual predator cues that are as similar to an actual encounter as possible. While robotic models have been shown to hold potential (*Krause, Winfield & Deneubourg, 2011*; *Ladu et al., 2015*; *Landgraf et al., 2016*; *Porfiri et al., 2019*), they may not always be easily accessible. We propose a technically simple method of simulating a predator attack, which can be produced with ease and in a short time. Like other video playback, also frontal playbacks of attacking predators may eventually lead to habituations though. However, this can effectively be prevented by running them in an unpredictable temporal scheme (M Watve, 2019, unpublished data).

Even though the focal fish interacted with live stimuli and frontal playbacks, the apparent attacks from the predators in either case did not elicit a fear response in the fish. One possible explanation is that the predators never actually managed to harm the focals even during a live presentation, which may have emboldened the focals.

*O'Connor et al. (2015)* previously found conspecific skin extract to reduce the response towards a visual predator cue in *N. pulcher*, which is opposite to the prediction that skin extracts represent and alarm cue and should increase anti-predator responses (see *Chivers & Smith, 1998* for comprehensive review). However, we did not find any such difference, neither when presenting skin extract alone or when paired with predator smell. Skin extract is assumed to contain alarm cues indicative of an injured conspecific, as shown in many other fish species including three-spine sticklebacks, *Gasterosteus aculeatus* (*Brown & Godin, 1997*) Rainbow trouts, *Onchoryhnchus mykiss* (*Brown & Smith, 1997*; *Brown*

& Smith, 1998), red swordtails Xiphophorus helleri (*Mirza, Scott & Chivers, 2001*) and rainbow darters, *Etheostoma caeruleum* (*Commens & Mathis, 1999*), although this is not always the case. A recent study showed that muscle tissue extracts induce antipredator responses more effectively than skin or other tissue extracts (*Meuthen et al., 2018*) in another African cichlid *P. taeniatus*. It is highly likely that skin extract does not contain alarm substances in *N. pulcher* either. Further investigation is required to determine whether and to what extent alarm cues are present in other tissues of *N. pulcher*. *Fischer et al. (2017b)* showed that both chemical and visual predator cues are used independently by *N. pulcher* in predator detection, but they did not administer them together. We administered chemical cues in combination with visual cues and our results suggest that *N. pulcher* ignore cues from predator odour or conspecific skin extract in the presence of visual cues. Similar results were seen in juvenile roach, *Rutilus rutilus*, which responded differently towards visual and chemical predator cues, but when presented with both cues together, the response was similar to visual cues alone (*Martin et al., 2010*).

Artificial proxies can help adhere to the 3Rs (Reduce, Refine, Replace) of ethical framework (*Russell & Burch, 1959*). While this is also an excellent way to increase standardization, it is important to make sure that the employed proxies indeed have the desired effect. For example, a species or population living under consistently high predation risk may ignore predator cues unless immediate threat is sensed (*Creel et al., 2008*; *Bell et al., 2009*). In such a case, it is important use a proxy that can simulate an immediate threat to induce measurable response in the prey.

## CONCLUSIONS

Here we show that *N. pulcher* do not use chemical cues from conspecific skin tissue as a indication of predation threat when combined with visual predator cues. The failure to respond to chemical cues may have been caused by precedence of visual cues over chemical cues when both cues in both modalities are presented together. We further show that including directed attacks by predator in a visual proxy induces responses similar to a live predator encounter. Most of previous work has focused on non-interactive predator visuals. We show that it is important to include prey-directed behaviour to simulate a threat level similar to that posed by a live predator.

## ACKNOWLEDGEMENTS

We would like to thank Evi Zwygart for logistic support. Andy Russell provided valuable comments on a previous draft of this manuscript.

### Funding

This study was funded by Swiss National Science Foundation grant (SNSF; grant 31003A_179208 to Barbara Taborsky). The funders had no role in study design, data collection and analysis, decision to publish, or preparation of the manuscript.

### Grant Disclosures

The following grant information was disclosed by the authors:
Swiss National Science Foundation: 31003A_179208.

### Competing Interests

The authors declare there are no competing interests.

### Author Contributions

- Mukta Watve conceived and designed the experiments, performed the experiments, analyzed the data, prepared figures and/or tables, authored or reviewed drafts of the paper, approved the final draft.
- Sebastian Prati performed the experiments, authored or reviewed drafts of the paper, approved the final draft.
- Barbara Taborsky conceived and designed the experiments, analyzed the data, contributed reagents/materials/analysis tools, authored or reviewed drafts of the paper, approved the final draft.

### Animal Ethics

The following information was supplied relating to ethical approvals (i.e., approving body and any reference numbers):

This study was conducted at the Ethologische Center Hasli, University of Bern, Switzerland under license BE-74/15, approved by the Veterinary Office of Kanton Bern, Switzerland.

### Data Availability

Raw data is available in the Supplemental File.

### Supplemental Information

Supplemental information for this article can be found online at http://dx.doi.org/10.7717/peerj.8149#supplemental-information.

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
