# Peer review of "Simulating more realistic predation threat using attack playbacks"

_PeerJ, doi:10.7717/peerj.8149_

## Round 0.1 · original submission · Minor Revisions

Two experts in your field have provided feedback on your article. While both praised your study, they both requested that you provide more detailed information about your protocol. I agree that this would enhance the clarity and reproducability of your study. As you prepare your revision, please respond carefully to each point raised by the reviewers.

Reviewer 1 ·

Basic reporting

No comment

Experimental design

No comment

Validity of the findings

No comment

Additional comments

The current manuscript examines the response of Neolamprologus pulcher to a variety of cues of predation threat which avoid risk of harm to predators or prey. Specifically, the authors examine the response of N. pulcher to their natural predator Lepidiolamprologus elongatus in an adjacent aquarium, video of an L. elongatus in a relaxed state or behaving aggressively. The authors found that compared to controls, the N. pulcher tended to respond most strongly to the live predator or the video playback of the predator behaving aggressively. In a second experiment, the authors expose N. pulcher to chemical cues of predation risk (L. elongatus holding tank water, conspecific skin extract, or a combination or the two) in conjunction with a video playback of a predator. The authors did not detect any effect of chemical cues on the response of the focal fish to the video. The paper is well written, and the experiments are well conducted. I do have some questions and suggestions for the authors which may help to refine the manuscript for publication.

Is it possible that aggressive behaviours directed towards a conspecific and predatory behaviours are different in form? If so, then the method of stimulus production may not capture prey directed behaviours as accurately as the authors assume. Furthermore, because the predators are not directing their attacks at the focal fish, but rather in their general direction (at simulated conspecific opponent), these videos may not fully simulate the presence of an actively hunting predator.

L117 – How were the videos edited down to remove gaps in the apparent predator attacks without creating unnatural jumps or cuts in the video?

L124 – It seems to me a better control would have been to edit a square over top of an actual elongatus on video. This would have created a stimulus in which the object moves like a real predator, but without the visual identity information. Having the shape move randomly changes both the appearance and the behaviour of the “predator”.

Perhaps with a larger testing tank more avoidance behaviour may have been observed?

L133 – Feeding the chemical cue donors flake food rather than N. pulcher may have muted the strength of this treatment, as aquatic prey species have been shown to pick up on predator diet cues in their excrement.

L156 – please provide more details of how being “oriented towards” or “attending to” the stimuli were scored.

Throughout, “chemical” would be more accurate than “olfactory” because these cues may also stimulate taste receptors, not just smell.

In figure 2, the overall trends seem to indicate greater interest in more complex or unpredictable stimuli, rather than increasing fear of increasingly more threatening stimuli.

Were any other measures of antipredator behaviour taken such as freezing, darting, thigmotaxis, increases or decreases in overall activity level. Especially with reference to the chemical stimuli, some of these other measures could reveal as yet inapparent effects.

Minor comments:
- L60: Given the lack of effect of chemical stimuli, perhaps it would be better to say “stimulate relevant sensory modalities”
- L68 & 245: trout
- L69: replace “acquire predator recognition” with “recognise predators”
- L110: presumably cm
- Please provide the full make and model of the tablet, along with its dimensions. It would also be useful to have some technical specifications such as the resolution, refresh rate, and colour gamut.
- L154: citation not in the reference list
- L188: more attentive
- The experiment numbers are reversed in the figure captions.

Reviewer 2 ·

Basic reporting

The manuscript entitled: Simulating more realistic predation threat using attack playbacks
investigated the effectiveness of olfactory and visual predator cues in Neolamprologus pulche, which suggested that playbacks of attacking predator can induce similar effect with a real predator.

Abstract
Line 25 Why do you emphasize this fish “cooperatively breeding”? It seems irrelevant with your topic.
Line 34 What do you mean by “prey-directed behaviours”? Please clarify it.

Introduction
Some sentence especially the introduction of knowledge gap seems lack of logic. And the description of knowledge gap seems speculative.
Line 42 The authors may need to explain why “such experiments are not always logistically”?
Line 71-73 This sentence seems speculative.
Line 89-91 This sentence is ambiguous, and else where. Because the visual cues were present during exp 2.
Methods
N. pulcher lives in groups in nature, but in your study, the behavior of N. pulcher was measured individually. Can you explain why?
Line 100 The N. pulcher was juvenile? Stock were caught from Lake Tanganyika? Did they have predation experience? And where were the predators come from?
Line 113, 124, 132 Please provide the body size of the predator and the prey. Otherwise, we will not sure whether the predator can act real threat on prey.
Line 113-116 Where did you put the camera? please provide more details. For the prey fish, it may have different angle of view on the predator between in the tank (real predator) and in the screen (playback). E.g. the distance between predator and prey, the size of the predator.
Line 113, 117 Please provide the detailed size of each tank which used in your study.
Line 114-117 It is a good method to obtain the “live stimulus” through mirror test, while it still has a flaw that the response may be different between attacking their own reflection and prey. Why didn’t you make the predator attack their real prey and film the playbacks? Some predators may not have mirror response and may restrict the utilization of your method.
Line 136 Why didn’t you kill the N. pulcher with euthanasia to avoid ethical issue?
Line 146-148 Did you use the same playback with exp 1? Why the playbacks “from 9 different L. elongatus individuals” in exp 2, but in exp 1, only two L. elongatus individuals were used? Too many objects may decrease standardization. Additionally, why did you use the lateral predator playbacks (swimming calmly) instead of frontal predator playbacks?

Results
Line 188 Which group compared to frontal playbacks?

Discussion:
Some contents are speculative and need more background and reference to support the results.
Line 230-231 see Results Line 191: No difference was found between lateral and frontal playbacks (p = 0.98; Table 2).
Line 236-238 Do you have some data or observation that whether the predators never managed to harm the prey fish? This kind of explain seems speculative and needs deeper discussion.
Line 238-239 It isn’t be a subsection.
Line 239-256 Some contents are speculative and need more background and reference. e.g. I suggest that you provide more background of N. pulcher’s habitats. N. pulcher is active only in the daytime? Or it’s habitat with high water transparency, thus usually use visual predator cues? Following review may be helpful to improve your discussion. Kelley J L, Magurran A E. Learned predator recognition and antipredator responses in fishes. Fish and Fisheries, 4, 2003, 4: 216-226.
Line 250-251 This sentence seems suspect. Intuitively, olfactory cues may be also important for prey to response to their predator.
Line 252-253 Fischer et al. (2017b) have showed that olfactory and visual predator cues are used independently by N. pulcher in predator detection, it was different with your study. You may need to explain why.
Line 254-256 This sentence seems ambiguous and speculative, because no significant difference was found, we don’t know whether visual cues or olfactory cues played a role.

Figures
Figure 1 The annotation in the plot was different from the caption. Predator tank was present only during exp 1? Or exp 2?
The note of “Tablet screen (Present only during trials)” is ambiguous, please clarify it.
Figure 2 Time in attention: It seems that no significant difference was exist between lateral and live, but there was a * in the plot.
For figure 2 and 3. The sequence number (e.g. a, b, c, d) for each plot was missing.

Experimental design

Manuscript describes original primary research. More details of the design and methods need to be provided.

Validity of the findings

Data are robust, but the figures need to be checked carefully.

Additional comments

The experiment appears well designed (although the authors need to provide some missing details about the protocol). Although the manuscript is fairly well written, there are parts where it is somewhat difficult to understand what the authors are trying to say, and it should be edited for clarity.

---

## Round 0.2 · accepted · Accept

The two reviewers who reviewed your original submission have now provided feedback on your revised article. Both noted that you had responded to all their concerns and both recommended publication. It is my pleasure, therefore, to accept your article for publication in PeerJ.

Reviewer 1 ·

Basic reporting

N/A

Experimental design

N/A

Validity of the findings

N/A

Additional comments

I think that the authors have done a satisfactory job of addressing my prior comments and I think the paper is now suitable for publication.

Note that the "olfactory" versus "chemical" terminology remains unchanged in the figures and captions and should be updated to match the main text.

Reviewer 2 ·

Basic reporting

The authors have thoroughly replied to my comments and have incorporated the required information in the text. Many imprecise sentences have been reformulated and their meaning is now clearer. The missing details I mentioned have been provided in this version.

Experimental design

Research question well defined.
Experimental design was appropriate to solve the authors' question.

Validity of the findings

Results can support their conclusions.

Additional comments

No comment.